# Near-complete chiral selection in rotational quantum states

JuHyeon Lee[1], Elahe Abdiha[1], Boris G. Sartakov[1], Gerard Meijer [1] &
Sandra Eibenberger-Arias [1] ✉

Controlling the internal quantum states of chiral molecules for a selected enantiomer has a wide range of fundamental applications from collision and reaction studies, quantum information to precision spectroscopy. Achieving full enantiomer-specific state transfer is a key requirement for such applications. Using tailored microwave fields, a chosen rotational state can be enriched for a selected enantiomer, even starting from a racemic mixture. This enables rapid switching between samples of different enantiomers in a given state, holding great promise, for instance, for measuring parity violation in chiral molecules. Although perfect state-specific enantiomeric enrichment is theoretically feasible, achieving the required experimental conditions seemed unrealistic. Here, we realize near-ideal conditions, overcoming both the limitations of thermal population and spatial degeneracy in rotational states. We achieve over 92% enantiomer-specific state transfer efficiency using enantio-pure samples. This indicates that 96% state-specific enantiomeric purity can be obtained from a racemic mixture, in an approach that is universally applicable to all chiral molecules of $C_1$ symmetry. Our work integrates the control over internal quantum states with molecular chirality, thus expanding the field of state-selective molecular beams studies to include chiral research.

Homochirality in living organisms, i.e. the preference for one enantiomer, is key to the origins of life, yet the exact mechanism behind it is still unclear. Among the various hypotheses[1–3], it is suggested that parity-violating energy differences between enantiomers, caused by the electro-weak force, can have initiated the chiral imbalance[4]. However, despite having been predicted for decades, parity violation in chiral molecules has not yet been experimentally observed[5–7].

Since Pasteur's discovery of optical activity, numerous chiral analysis techniques, such as optical rotation, circular dichroism, and Raman optical activity have been developed[8–11]. These traditional methods inherently produce weak signals due to their reliance on weak interactions of the sample with the magnetic field of the light[12]. Recently, research on chiral molecules has been rejuvenated by the emergence of new types of spectroscopic methods that rely exclusively on strong electric-dipole interactions. These include Coulomb explosion imaging[13], photo-electron circular dichroism[14,15], and microwave three-wave

mixing[16,17]. These methods are well-suited for studies on dilute samples as they offer remarkably strong enantiomer-sensitive responses.

Enantiomer-specific state transfer (ESST) is a particularly intriguing extension of microwave three-wave mixing. Beyond chiral analysis, ESST enables enantiomer-specific control over the population in rotational states[18,19]. While ESST has thus far experimentally only been demonstrated for rotational states, theoretically, it can be extended to vibrational and electronic degrees of freedom[20,21]. ESST utilizes a unique spectroscopic feature of chiral molecules of $C_1$ symmetry, i.e., molecules that have no symmetry operation other than the identity. These molecules possess closed triads of electric-dipole-allowed transitions between rotational states[22], which is the key property that enables differentiation between enantiomers. The enantiomer-selectivity of ESST is mathematically governed by the triple product of the three components of the electric-dipole moment, that has opposite signs for different enantiomers. In principle, when only a

[1]Fritz-Haber-Institut der Max-Planck-Gesellschaft, Berlin 14195, Germany. ✉e-mail: eibenberger@fhi-berlin.mpg.de

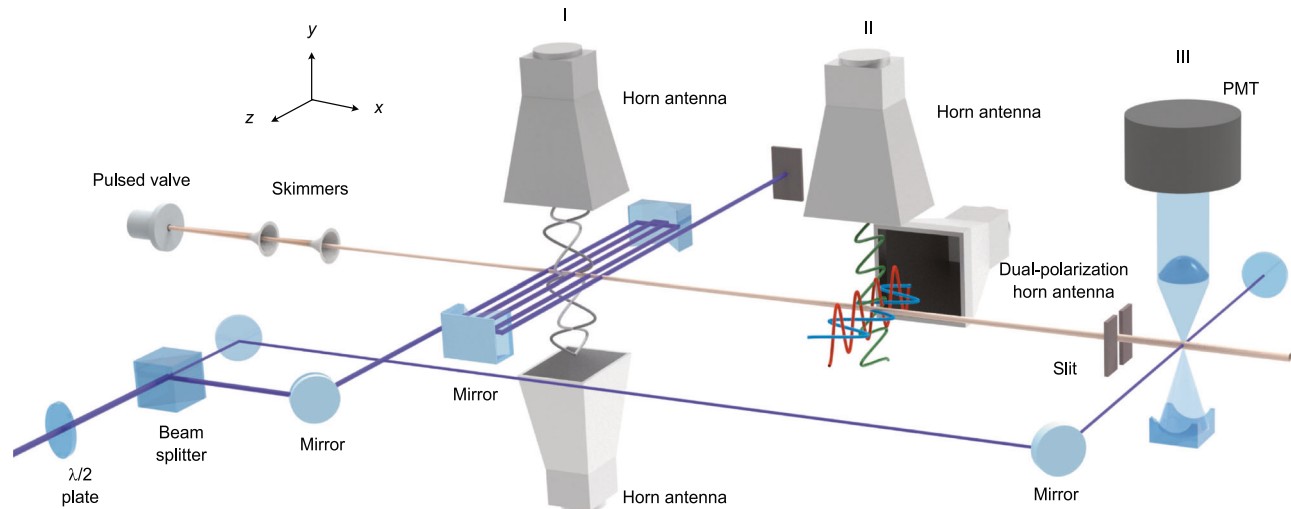

**Fig. 1 | Scheme of the experimental setup.** Jet-cooled 1-indanol passes through two skimmers and then traverses three distinct regions (I-III). In (I), the molecules interact with both the UV depletion laser and with MW fields that drive the *b*-type transition. The optical multi-pass setup extends the interaction time to ~30 μs. In (II), the molecules interact with a sequence of three orthogonally, linearly polarized MW fields. In (III), the target rotational state is probed using the same UV laser as used for depletion. The laser-induced fluorescence is detected using a photo-multiplier tube (PMT).

single state is initially populated and when the spatial degeneracy is properly addressed, ESST can reach 100% efficiency.

In the ideal scenario, where only one state is populated, and there is a single Rabi frequency for each of the three transitions, perfect transfer efficiency can be achieved using a $\pi/2 - \pi - \pi/2$ pulse sequence[23,24]. In this sequence, the $\pi/2$ pulses generate maximum coherence between states and the $\pi$ pulse exchanges population between states. However, early ESST studies reported only modest state-specific enantiomeric enrichment, limited to a few percent[18,19]. This is primarily due to the thermal population of rotational states[25], which prevails even at rotational temperatures of around 1 K that can be achieved in molecular beams, for instance. In addition, the spatial degeneracy of these states often results in multiple Rabi frequencies for each of the three transitions[23,26]. To mitigate the effect of thermal population, ultraviolet (UV) or additional microwave (MW) radiation has been used to deplete one of the rotational states before the ESST process[27–29], thereby significantly enhancing the transfer efficiency. The issue of multiple Rabi frequencies due to spatial degeneracy is circumvented when targeting a triad of rotational states that includes the absolute rotational ground state[30]. In this way it has been possible to perform the first quantitative study of ESST[27], albeit under conditions that were not yet ideal.

The present study reports on the experimental realization of the ideal scenario for enantiomer-specific state transfer. By employing MW-UV double resonance in the depletion setup, the thermal population from two rotational states of the triad is removed prior to ESST. Full state-specific enantiomeric enrichment can be obtained by applying this two-level depletion scheme to a triad of rotational states that includes the rotational ground state. While this experimental approach may not produce bulk amounts of pure enantiomers, it is very valuable for fundamental science studies on chiral molecules. Despite only a small fraction of molecules being in a given quantum state, these can be selectively detected with a high sensitivity.

## Results

The experimental setup is schematically depicted in Fig. 1. The chiral molecule 1-indanol is heated to ~80 °C and seeded into neon. The gas mixture is expanded at ~2 bar backing pressure out of a pulsed valve, operated at 30 Hz, into a vacuum. The thus produced jet-cooled molecular beam has a rotational temperature of ~1 K and a speed of ~800 m/s. The molecular beam is collimated by two skimmers of

3 mm and 1 mm diameter opening, and subsequently passes through the depletion region (I), the ESST region (II), and the detection region (III), as outlined below. The relevant rotational states and MW frequencies of 1-indanol are depicted in Fig. 2a. Details of the excitation schemes at the interaction regions are illustrated in Fig. 2b and c. As indicated in these figures, states $|1_{01}\rangle$ and $|1_{10}\rangle$ are depleted in region (I), population is transferred between all three states in region (II), and the population in the target state $|1_{01}\rangle$ is monitored in region (III).

In the depletion region (I), about 40 mW of tunable continuous-wave UV laser radiation with a bandwidth of less than 1 MHz crosses the molecular beam perpendicularly in a multi-pass arrangement. In this way, the total interaction time of the molecules with the UV radiation is extended. The laser frequency is set to selectively excite from the target rotational state $|1_{01}\rangle$ to the electronically excited state on the $S_1(2_{02}) \leftarrow S_0(1_{01})$ R-branch line[31], addressing all $M_J$ sub-levels. Once excited, the molecules rapidly radiate, predominantly to higher vibrational states and other rotational states within the $S_0$ electronic state[32]. For the small fraction of molecules that radiates back to the original state, the process is repeated, effectively depleting the $|1_{01}\rangle$ state. Simultaneously, MW fields that drive the *b*-type transition are applied to connect state $|1_{10}\rangle$ to the target state $|1_{01}\rangle$, thereby depleting both levels via optical pumping. The polarization of these MW fields is switched back and forth between the **z**- and **x**-, which enables coupling all $M_J$ sub-levels of the rotational states. This is crucial for complete depletion, as otherwise, according to the selection rules, the $M_J = 0$ sub-level in state $|1_{10}\rangle$ would remain populated. In Fig. 2b it is shown, how the different $M_J$ sub-levels are addressed by each MW polarization direction. Further details are given in the Methods.

In the ESST region (II), a sequence of three resonant, linearly polarized, and mutually orthogonally polarized MW fields is applied consecutively in the following order: $|1_{01}\rangle \overset{\pi/2,\phi_a}{\leftarrow} |0_{00}\rangle \overset{\pi,\phi_c}{\rightarrow} |1_{10}\rangle \overset{\pi/2,\phi_b}{\rightarrow} |1_{01}\rangle$, where $\phi_i$ represents the phase of the MW field driving the *i*-type transition, with $i = a,b$ or $c$. Note that other pulse sequences can be used for ESST if the first MW pulse drives a transition from the initially populated rotational state. Enantiomer-specific state transfer is then achieved between the two rotational states connected by the final MW pulse. In our chosen sequence, enantiomeric enrichment is realized in two excited rotational states, but we only probe the population in the target rotational state. The enantiomer-selectivity of ESST is determined by the relative phases of

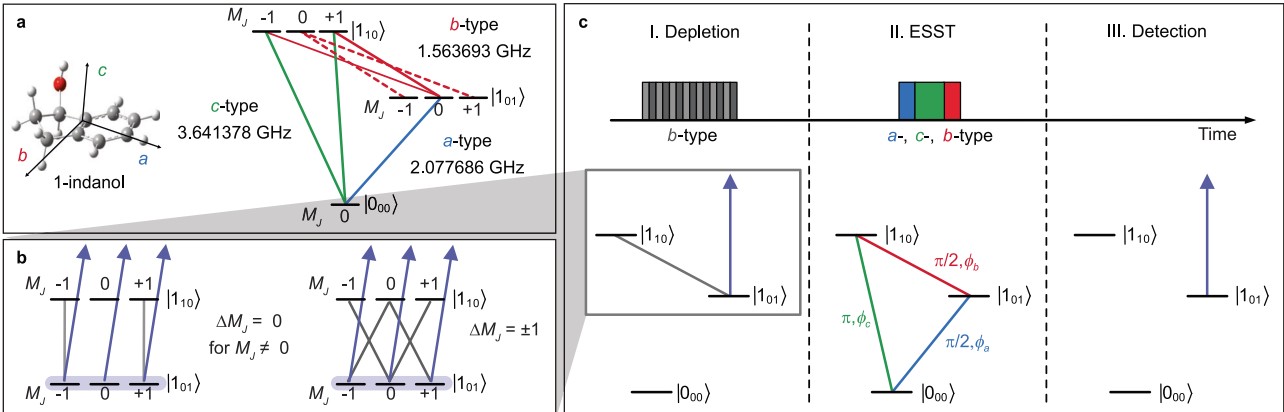

**Fig. 2 | Schemes of the experimental procedure using 1-indanol. a** Illustration of the triad of rotational states of 1-indanol used in this study, shown in standard spectroscopic notation $\left|J_{K_aK_c}\right\rangle$. All $M_J$ sub-levels and allowed transitions are represented, with frequencies and types of MW transitions marked. **b** Depiction of depletion schemes using MW-UV double resonance. Two scenarios are presented: using polarization along **z** (left) and along **x** (right), with the respective selection rules. **c** Time sequences of the applied MW fields (top) and the excitation schemes applied at each stage of the experiment (bottom).

the MW fields. In the experiment, the phase of the final pulse $\phi_b$ is scanned in 20° increments from 0° to 720°, while the phases of the first two MW fields, $\phi_a$ and $\phi_c$, are kept fixed.

In the detection region (III), the same UV laser as used in the depletion region probes the population in the target rotational state, utilizing only about 10% of the laser power to avoid line broadening. The laser is aligned perpendicularly to the molecular beam and parallel to the laser in the depletion region, thus interacting with the same group of molecules as in the depletion region. The laser-induced fluorescence (LIF) emitted by the molecules is detected using a photomultiplier tube (PMT).

In this experimental approach, the ESST signal for a given enantiomer, defined as the population in the target state $\left|1_{01}\right\rangle$ at the end of the ESST process, is given by the following expression[29]:

$$\frac{1}{2}\left[n_{0_{00}} + 4n_{1_{01}} + n_{1_{10}} \pm \left(n_{0_{00}} - n_{1_{01}}\right)\sin\left(\phi_a - \phi_c + \phi_b\right)\right] \quad (1)$$

where $n_{0_{00}}$, $n_{1_{01}}$, and $n_{1_{10}}$ is the initial population of each $M_J$ sub-level of the states $\left|0_{00}\right\rangle$, $\left|1_{01}\right\rangle$, and $\left|1_{10}\right\rangle$, respectively, at the beginning of the ESST process and where the $\pm$ sign is used for different enantiomers. Here, the handedness of the coordinate system dictates the handedness of the molecules associated with the $\pm$ sign. The amplitude-to-mean ratio of this expression is the measure for maximum state-specific enantiomeric enrichment, which refers to the enantiomeric excess achievable in a chosen rotational state when starting from a racemic mixture. For example, an 80% state-specific enantiomeric enrichment means that, starting from a racemic mixture, 80% of the molecules in the target rotational state are one enantiomer, while 20% are the racemic mixture. In other words, this corresponds to a composition of 90% of one enantiomer and 10% of the other enantiomer in the target state. Achieving 100% state-specific enantiomeric enrichment requires that both upper levels are initially empty, i.e., $n_{1_{01}} = n_{1_{10}} = 0$. It is seen from this expression how any remaining population in states $\left|1_{01}\right\rangle$ and $\left|1_{10}\right\rangle$ adversely affects the state-specific enantiomeric enrichment. Moreover, the population in the target state impacts the mean of the ESST signal four times more than the population in state $\left|1_{10}\right\rangle$.

Measurements are performed using commercially available, enantiopure samples of 1-indanol. The enantiopurity for both samples is better than 99.8% as determined by chiral high-performance liquid chromatography. Separate molecular beam sources are used for the two enantiomers to avoid any cross-contamination. To ensure an

optimal $\pi/2 - \pi - \pi/2$ pulse sequence, Rabi oscillation curves for each MW transition are measured prior to ESST (see Suppl. Note 3). The $\pi$-pulse conditions are determined from the pulse durations that correspond to the first maxima of these curves, as indicated by vertical bars in Fig. 3a. Due to the use of different molecular beam sources, the $\pi$-pulse durations are slightly different for the measurements on the (R)- and (S)-enantiomer.

Figure 3b shows the ESST signals, normalized to the thermal population in the target state $\left|1_{01}\right\rangle$, as a function of the relative MW phase. The signals for the (R)- and (S)-enantiomers are shown in black and red, respectively, with error bars representing the standard errors. The mean and amplitude values from sinusoidal fits to the data are given. These values yield a maximum state-specific enantiomeric enrichment of 90.2(1.9)% for the (R)-enantiomer and 92.4(2.1)% for the (S)-enantiomer. This latter value means that when starting from a racemic mixture, we can selectively obtain 92% of one enantiomer in the target rotational state, with the remaining 8% being racemic. Consequently, it results in a final composition where the target state comprises 96% of one enantiomer and 4% of the other. The high degree of state-specific enantiomeric purity is best seen by the proximity of the minima of the sine curve to zero, and two relevant segments are therefore shown enlarged in Fig. 3b.

The two ESST curves in Fig. 3b are shown with a phase difference of exactly 180°. When using identical pulse durations for the (R)- and (S)-enantiomer, their ESST curves are indeed confirmed to be 180° out-of-phase (see Suppl. Note 5). However, optimal transfer efficiency for both enantiomers is only obtained when using pulse durations that are optimized for each enantiomer individually. This results in an additional phase offset between the measurements, that has been corrected for in Fig. 3b.

In the experiments, ~2% of the original thermal populations in the states $\left|1_{01}\right\rangle$ and $\left|1_{10}\right\rangle$ is measured to be present in the detection region (see Methods). This population is attributed to re-filling as a result of in-beam collisions with the carrier gas atoms while the molecules travel from the depletion region to the detection region[27,29]. This is confirmed by the observation that reducing the carrier gas density by installing a second skimmer is beneficial, i.e., that it reduces this population. When depleting just in front of the LIF detection region, basically no population is measured to be present. Using this information in a model as described in the Suppl. Note 2, the population in both of these states is estimated to be about 1.4% upon entering the ESST region.

Calculated normalized ESST curves using expression (1) and assuming thermal populations at 1.1 K for (R)-indanol and 0.7 K for (S)-

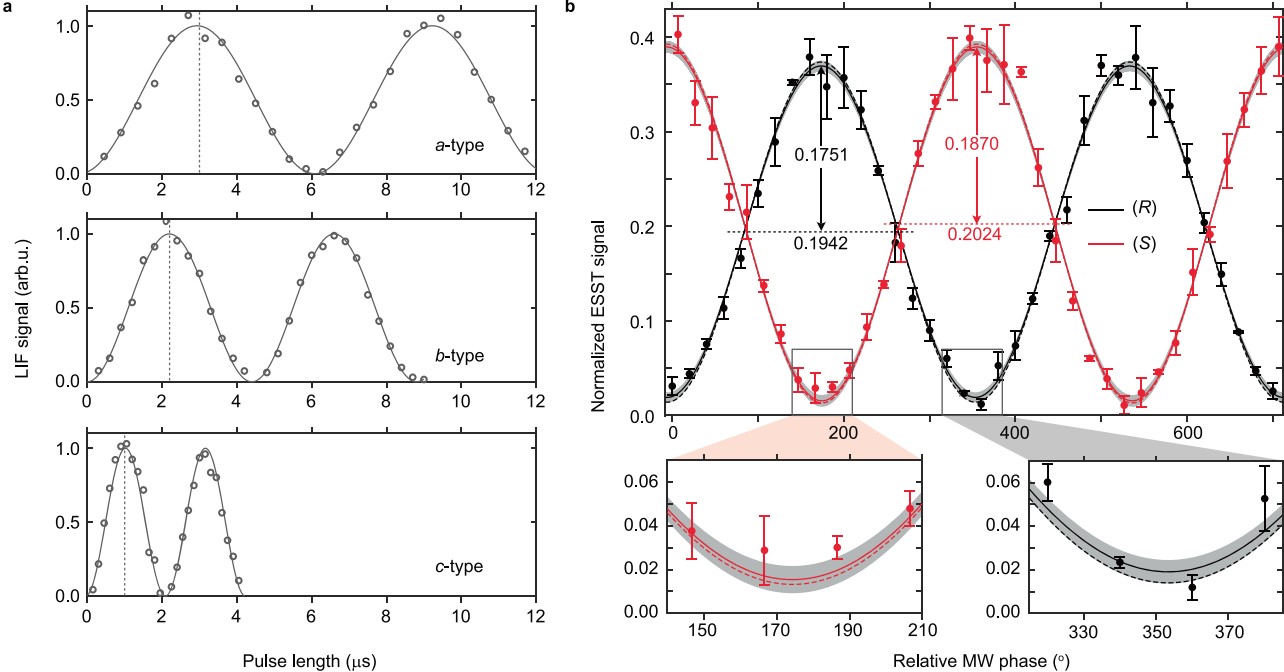

**Fig. 3 | Rabi oscillation curves and ESST results. a** Rabi oscillation curves of the MW transitions are shown for (R)−1-indanol. The π-pulse durations are marked by dashed vertical lines. **b** Normalized ESST signal is shown as a function of the relative MW phase for (R)- and (S)-enantiomers in black and red, respectively. The standard error on each measurement point is indicated by error bars. Calculated normalized ESST curves using expression (1) are shown with dashed lines. The gray shaded areas around the experimental curves show the standard deviation from the sine fit. Two regions of specific interest are shown enlarged in the boxes below.

indanol, are shown in Fig. 3b (dashed). These calculations incorporate the 1.4% population in the two upper levels as well as the 0.2% enantiomer-impurity. The calculated amplitude and mean of the normalized ESST signal agree very well with those of the experiment, for both enantiomers, indicating that enantiopure substances are not required to determine the absolute enantiomeric excess of a sample. Moreover, with the absolute phases of the three MW fields known, it is possible to determine the absolute configuration, even within a racemic mixture. This sets our approach apart from most other chiral discrimination techniques.

## Discussion
We achieve near-ideal experimental conditions for ESST by implementing a two-level depletion scheme, resulting in unprecedentedly high state-specific enantiomeric enrichment of 92%. The data presented here show near-complete quantum state control of the chiral molecule 1-indanol to the extent that more than 96% enantiomer-selectivity can be obtained when starting from a racemic mixture, which can be verified following a scheme presented elsewhere[33]. Currently, the only experimental limitation from reaching 100% enantiomer-selectivity stems from in-beam collisions, which can be avoided by reducing the distances between the three interaction regions. The experimental approach outlined here is universally applicable to the large majority of chiral molecules. If the chiral molecule under study does not possess a chromophore, then full population transfer to an excited vibrational state can be used, for example, as rotational state selectivity is the only requirement for depletion and detection[34]. Our approach of controlling the internal quantum states of chiral molecules for a selected enantiomer offers a wide range of fundamental applications[35–38]. With its capability to create enantiopure quantum states starting from a racemic mixture, this approach has the potential to significantly advance the experimental methods to measure parity-violation effects in chiral molecules[39]. Moreover, by incorporating state-dependent deflection or focusing schemes[40], these measurements pave the way for spatial separation of chiral molecules in the gas phase.

## Methods
### Two-level depletion scheme
In the depletion region (I), the populations in states $|1_{01}\rangle$ and $|1_{10}\rangle$ are removed via MW-UV double resonance. To ensure complete depletion of both states, it is crucial for the microwaves to transfer the population from all $M_J$ sub-levels of state $|1_{10}\rangle$ to state $|1_{01}\rangle$. To achieve this, the polarization of the MW radiation is switched back and forth between the **z**- and **x**-axes. The first MW polarization direction sets the quantization axis along the **z**-axis, enabling $\Delta M_J = 0$ transitions from $M_J = \pm 1$ but not from $M_J = 0$, due to selection rules, as illustrated at the top of Supplementary Fig. 1a. The MW field polarized along the **x**-axis, enables $\Delta M_J = \pm 1$ transitions, thereby coupling all $M_J$ sub-levels as depicted at the bottom of Supplementary Fig. 1a.

In the experiment, a switch is used to toggle the microwaves between two MW horn antennas, each of which has a different polarization direction, as depicted in Supplementary Fig. 1b. The switching is controlled by the voltage applied to the driver: above 2.5 V, it connects to the antenna broadcasting a MW field polarized along the **z**-axis, and below 2.5 V, it connects to the antenna broadcasting a MW field polarized along the **x**-axis. For toggling between the antennas, a square wave burst (between 0 and 3 V) from a function generator is applied to the switch driver. The switching period is set to be 6 µs based on the measured π-pulse duration of the MW field that drives the b-type transition.

### Laser alignment
In the detection region (III), the same UV laser as used in the depletion region probes the population in the target rotational state, utilizing only about 10% of the laser power to avoid line broadening. To ensure a perpendicular intersection of the laser with the molecular beam, the light is retro-reflected and aligned as to minimize the width of the

spectral line. Care is taken, that the laser beam is also aligned parallel to the laser beam in the depletion region to ascertain that in both regions, the same transverse velocity group of molecules is interacted with. The probing transition has a natural linewidth of approximately 5 MHz[31], corresponding to a width of the transverse velocity distribution of ∼1.4 m/s.

### Interpreting the LIF signals

In Supplementary Fig. 2, the measured LIF signal probing the population in the target rotational state $|1_{01}\rangle$ is shown for three different scenarios. The LIF signal, recorded without the UV laser or MW fields in the depletion region (I), shown in black, represents the original thermal population of the target state. The LIF signal, recorded with depletion using the UV laser, shown in purple, represents the population present in the target state after depletion. To fully deplete both states $|1_{01}\rangle$ and $|1_{10}\rangle$, MW fields, as well as the UV laser are utilized in the depletion region. The LIF signal, recorded after a π-pulse population swap has been applied between the states $|1_{01}\rangle$ and $|1_{10}\rangle$ in the ESST region (II), shown in gray, represents the population present in state $|1_{10}\rangle$ after depletion.

In the current experimental setup, it is observed that initially depleted states are repopulated as the molecules travel from the depletion region to the detection region. Supplementary Fig. 2 shows that about 2% of the original thermal population for both states is present in the detection region. The re-filling process is caused by in-beam collisions of the molecules with carrier gas atoms[27,29]. This is confirmed by the observation that reducing the carrier gas density by installing a second skimmer is beneficial. When depleting just in front of the LIF detection region, basically no population is measured to remain in the state. This allows for the assumption that depleted states are completely empty in the depletion region.

## Data availability

The raw data and the source data used in this study are available in the Edmond database (https://doi.org/10.17617/3.C08CCR)[41]. The source data are also provided with this paper. Source data are provided with this paper.

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

## Acknowledgements

We are grateful to Johannes Bischoff and Alicia Hernandez-Castillo for their contributions to the earlier stages of the experiment. We thank Marco De Pas, Sebastian Kray, Henrik Haak, Daniel Fontoura Barroso, and Russell Thomas, as well as the teams of the mechanical and electronics workshop of the Fritz Haber Institute for excellent technical and laser support. Funded/Co-funded by the European Union (ERC, COCOCIMO, 101116866). Views and opinions expressed are, however, those of the author(s) only and do not necessarily reflect those of the European Union or the European Research Council. Neither the European Union nor the granting authority can be held responsible for them.

## Author contributions

S.E.-A. initiated the research direction. S.E.-A. and G.M. supervised the project. J.H.L. and E.A. performed the experiment and analyzed the data. B.G.S. provided the theoretical background. J.H.L. drafted the initial manuscript. All authors actively participated in discussing the results and contributed to refining the final manuscript.

## Funding

## Competing interests

The authors declare no competing interests.
