## [Peer Review File · Nature Communications]

Near-complete chiral selection in rotational quantum statesReviewer #1 (Remarks to the Author):

In this contribution Lee¹, Abdiha, Sartakov, Meijer and Eibenberger-Arias show that they can nearly completely enantioselectively and quantitatively excite one of the enantiomers of 1-indanol to a particular excited state. This achievement is indeed novel and of great interest and importance to a wide community of chemists and physicists.

The manuscript is written in a concise and clear manner.

Although my overall impression of the work is favourable, I would like to seek further clarification from the authors regarding the following two points the

1) As a relative outsider, not being involved with this kind of high order microwave spectroscopy, it seems that in general one would want to reach the following two goals (a and b) :

a) Achieve a high enantiomeric purity in the target excited state: of all molecules in the particular target excited state, close to 100% should be of a particular handedness

b) quantitative transfer of the population of one particular enantiomer from the ground state to the target excited state: of all molecules in the probe volume, close to a 100% of all molecules with a particular handedness, say R, should be in the target excited state. This second target, introduces a 'novel' additional kind of enantiomeric excited state purity parameter that is different from what chemists and spectroscopists are used to.

Now my first question to the authors is how they have actually defined their 'enantiomeric enrichment'? Here I refer directly to what seems to be the most important sentence in the paper "These values yield a maximum state-specific enantiomeric of 90.2(1.9)% for the (R)-enantiomer and 92.4(2.1)% for the (S)-enantiomer." on page 9

Is this refer the enrichment in one enantiomer relative to all molecules in that excited target state or is it the enrichment relative to all molecule in the probe volume, irrespective of their state ?

In addition, I have difficulty reproducing the numbers stated for the enantiomeric enrichment ("90.2(1.9)% for the (R)-enantiomer and 92.4(2.1)% for the (S)-enantiomer." on page 9) from the data in the Figure 3b

I suggest that in the supporting information the authors detail their calculation of the enrichment

2) The experiments performed in this study seem to be done only on enantiomerically pure samples of the R and S antipodes of the indanol.

Now in an actual experiment on a 50-50% mixture of R and S molecules, transfer of excitation energy between R and S molecules seems theoretically possible. Such energy transfer would lower the enantiomeric enrichment that can be reached in the actual racemic mixture. Could the authors comment on the likelihood of such energy transfer under the conditions of the experiment ?

Reviewer #2 (Remarks to the Author):

The paper entitled "Full quantum state control of chiral molecules" reports on enantiomer-specific state transfer (ESST) experiments to achieve almost complete enantio-enrichment in a rotational state using rotational spectroscopy combined with LIF. The main result of the current study with respect to previous works by the authors and others is the increase in the enantiopurity of the rotational state, reporting in this case up to a 96%. The authors successfully implemented a depletion scheme including MW-UV double resonance that allows to removal of the thermal population before performing ESST experiments with MW fields. The experiments are competently performed, and the results are sound and interesting. The paper is clearly written and reports the experimental finding in a clear manner. However, I consider that they are a natural extension of

previous research and very limited to a very reduced pool of chiral molecules. This is because, the current system fulfills some of the basic requirements that won't be easily found in most chiral molecules, such as having a chromophore group and the ability to work on the rotational ground state 000. As stated by the authors, the use of IR laser could help circumvent the first requirement as most chiral molecules do not have a chromophore group, but even in this case vibrational to rotational energy transfer will potentially affect achievable enrichment. The ability to work on the rotational ground state is a major limitation. This is because the required MW or RF frequencies connected to the ground state are largely affected by the size and geometry of the system under study, making it in some cases hard to reach. Without fulfillment of this, the M-degeneracy will impact the state transfer, limiting thus the enantiopurity of the target rotational level. With this being said, I am hesitant to recommend this paper for publication in Nature Communications. I believe that the novelty of the current study is mostly incremental with respect to the author's previous work (PHYSICAL REVIEW LETTERS 128, 173001 (2022)), and the general applicability to most chiral molecules is rather limited. Despite the undeniable quality of the work, I find it more appropriate for a more specialized journal.

Nevertheless, I have several comments that the authors might want to consider:

1. I find the figures to be clear and very informative. I recommend reporting the frequencies in GHz to improve readability. Also, in the caption of Figure 2 the notation is JKaKc.
2. Related to this, I was not able to find some important specifications of the amplifiers such as frequency coverage and output power. This info is given for the laser.
3. There is very little to no information about the generation and control of the pulses during the ESST stage. In a paper presenting an experimental setup, I think this information is relevant and should be included in the methods or SI.
4. I believe that the section about the rabi frequencies (Rabi flip angles in reality) is introduced without context. A brief introduction or even a small scheme would help the non-specialist follow the reasoning better.
5. The authors state that this experiment can be performed in a racemic mixture, however, the authors only report results using enantiopure samples. It would be valuable to show the results using a racemic mixture. This would also avoid using different excitation pulses that complicate the experiment. Under normal conditions, both enantiomers should exhibit the same optimal conditions.
6. Related to this, it is unclear why different rotational temperatures were used to calculate normalized ESST curves for each enantiomer. This needs to be further clarified.
7. Lastly, have the authors considered dephasing from the ESST region to the detection? This could also affect enantiomer enrichment. This should be dependent on the distance between the ESST region and detection.

Reviewer #3 (Remarks to the Author):

See attached for formatted report.

Review of manuscript NCOMMS-24-25450-T "Full quantum state control of chiral molecules"

This manuscript describes an optimized version of the experiment first reported in Lee et al.

(10.1103/PhysRevLett.128.173001). The new experiment improves the initial depletion step of the previous experiment, by adding a microwave field to resonantly couple the 110 and 101 states.

The field is switched between two orthogonal polarizations to transfer population from all spatially degenerate levels of 110 to the 101 level where population is optically pumped to an excited electronic state and effectively "removed". This preparation leaves population only in the initial 000 level of the three-level experimental system. A thus prepared sample of either (S)- or (R)-1-indanol seeded in a neon molecular beam is subjected to the standard three pulse sequence for enantiomer-selective state transfer (ESST) to the final 101 level. Detection is performed by LIF of the same electronic transition used in the depletion step. With optimized π or $\pi/2$ pulses on the ESST step, the authors demonstrate 90% or better enantiomeric enrichment on the basis of the contrast (amplitude/mean ratio) obtained by varying the phase of the final microwave pulse. A phase shift of 180° is observed between the (R)- and (S)-enantiomers as expected and agreement with the theoretical model is excellent. The residual racemic population of the final level is attributed primarily to re-population of the depleted 101 and 110 by in-beam collisions, and experimental evidence and modelling demonstrate that this mechanism is operative. The authors

point out that if their technique were to be applied to a racemic mixture, the target 101 state would be composed of up to 96% a single enantiomer. The utility of such enantiopure state preparation for chemical physics and investigations of parity violation in chiral molecules is significant.

I recommend this manuscript for publication after consideration of the minor comments below. The manuscript is well written and contains easily interpretable figures, and the study itself is well-designed, and carefully executed and analyzed. The results represent a major advance in the manipulation of chiral molecules with far-reaching implications. It will be of wide and significant interest for the readership of Nature Communications.

Page 5 Line 22, Page 6 Line 1, Fig 2, Page 14 Lines 7 & 9

In several instances, a quantum number MZ is referred to. I understand this to mean the space-fixed projection of the total angular momentum along the Z-axis. However, the authors also use the label MJ which typically refers to the space-fixed projection of the total angular momentum along the quantization axis, which is typically the Z-axis by convention and is specifically the Z-axis in this experiment. Thus, these two labels to my understanding refer to the same quantity. Perhaps I am being over-simplistic and the authors feel that making the distinction between MJ and MZ is especially important. If that is the case, I would make the difference between the two labels as transparent as possible.

Page 10, Lines 8-10

From Equation 1, I see that the signal is controlled by the phases of the three MW fields. Is the \pm distinction known a priori for any pair of enantiomers or can be reliably calculated? I believe that quantum chemistry calculations provide good predictions – this might be specified as an additional requirement. For my own understanding, is it correct that what distinguishes this method from the MW three-wave mixing technique is that the precise phase relationship of three applied fields must be known here, rather than two applied and one detected field (much harder) in the case of MW3M?

General

It appears that one significant limitation in implementing this approach with a racemic mixture is that, unless the enantiomeric excess of the sample is previously known, the enantiomeric state purity of the racemate must be inferred from an enantiopure study. However, as the authors showed, even subtle differences in the nozzle/source conditions can impact the optimal parameters for ESST. A small change in the optimal relative MW phase and pulse length may lead to a different enantiopurity than what would have been inferred from an enantiopure experiment with optimal parameters. Can the authors comment on this limitation? Are there approaches to verify the final enantiopurity through a measurement? I know that in other experiments on enantiomer-selective population enrichment, the final state purity can be measured by MW3M, however that would not seem to be compatible with this experiment.

Reviewer #3 Attachment on the following page

Review of manuscript NCOMMS-24-25450-T “Full quantum state control of chiral molecules”

This manuscript describes an optimized version of the experiment first reported in Lee et al. (10.1103/PhysRevLett.128.173001). The new experiment improves the initial depletion step of the previous experiment, by adding a microwave field to resonantly couple the 1_{10} and 1_{01} states. The field is switched between two orthogonal polarizations to transfer population from all spatially degenerate levels of 1_{10} to the 1_{01} level where population is optically pumped to an excited electronic state and effectively “removed”. This preparation leaves population only in the initial 0_{00} level of the three-level experimental system. A thus prepared sample of either (S)- or (R)-1-indanol seeded in a neon molecular beam is subjected to the standard three pulse sequence for enantiomer-selective state transfer (ESST) to the final 1_{01} level. Detection is performed by LIF of the same electronic transition used in the depletion step. With optimized π or $\pi/2$ pulses on the ESST step, the authors demonstrate 90% or better enantiomeric enrichment on the basis of the contrast (amplitude/mean ratio) obtained by varying the phase of the final microwave pulse. A phase shift of 180° is observed between the (R)- and (S)-enantiomers as expected and agreement with the theoretical model is excellent. The residual racemic population of the final level is attributed primarily to re-population of the depleted 1_{01} and 1_{10} by in-beam collisions, and experimental evidence and modelling demonstrate that this mechanism is operative. The authors point out that if their technique were to be applied to a racemic mixture, the target 1_{01} state would be composed of up to 96% a single enantiomer. The utility of such enantiopure state preparation for chemical physics and investigations of parity violation in chiral molecules is significant.

I recommend this manuscript for publication after consideration of the minor comments below. The manuscript is well written and contains easily interpretable figures, and the study itself is well-designed, and carefully executed and analyzed. The results represent a major advance in the manipulation of chiral molecules with far-reaching implications. It will be of wide and significant interest for the readership of Nature Communications.

Page 5 Line 22, Page 6 Line 1, Fig 2, Page 14 Lines 7 & 9

In several instances, a quantum number M_Z is referred to. I understand this to mean the space-fixed projection of the total angular momentum along the Z-axis. However, the authors also use

the label M_I which typically refers to the space-fixed projection of the total angular momentum along the quantization axis, which is typically the Z-axis by convention and is specifically the Z-axis in this experiment. Thus, these two labels to my understanding refer to the same quantity. Perhaps I am being over-simplistic and the authors feel that making the distinction between M_I and M_Z is especially important. If that is the case, I would make the difference between the two labels as transparent as possible.

Page 10, Lines 8-10

From Equation 1, I see that the signal is controlled by the phases of the three MW fields. Is the \pm distinction known *a priori* for any pair of enantiomers or can be reliably calculated? I believe that quantum chemistry calculations provide good predictions – this might be specified as an additional requirement. For my own understanding, is it correct that what distinguishes this method from the MW three-wave mixing technique is that the precise phase relationship of three *applied* fields must be known here, rather than two applied and one *detected* field (much harder) in the case of MW3M?

General

It appears that one significant limitation in implementing this approach with a racemic mixture is that, unless the enantiomeric excess of the sample is previously known, the enantiomeric state purity of the racemate must be inferred from an enantiopure study. However, as the authors showed, even subtle differences in the nozzle/source conditions can impact the optimal parameters for ESST. A small change in the optimal relative MW phase and pulse length may lead to a different enantiopurity than what would have been inferred from an enantiopure experiment with optimal parameters. Can the authors comment on this limitation? Are there approaches to verify the final enantiopurity through a measurement? I know that in other experiments on enantiomer-selective population enrichment, the final state purity can be measured by MW3M, however that would not seem to be compatible with this experiment.

Reviewer #4 (Remarks to the Author):

The authors report a new demonstration of Enantiomer-Specific State Transfer, which yields a high measured transfer yield, due to careful control of microwave fields, and (critically) depletion of several states via optical pumping. Using their own definitions of fidelity, the authors demonstrate the highest ESST to date, at 96 percent. The remaining 4 percent come from small technical imperfections, such as residual collisions within the beam.

The work is carefully done, and the 96 percent purity is convincingly demonstrated. In fact, the abstract accurately summarizes what has been done by the authors. The title does not.

The title of the paper, "Full quantum state control of chiral molecules," is misleading. Full quantum state control of ANY polyatomic molecule - let alone a chiral one - would be an impressive achievement, well beyond the state of the art. This has not been achieved here. The molecules in the beam start in a thermal distribution of states, at 1.1 K, and none of the steps taken - optical pumping, or reversible microwave pulses - significantly reduce the entropy of this beam. "Full quantum state control" would mean, at a minimum, that the molecules are put in a single quantum state, which is not the case. In addition, the hyperfine states of the molecules are not addressed in any form. A more accurate title would be something like "near unity enantiomer-specific state purity from a racemic sample. This title obviously carries less punch - but it reflects the work performed, which the current title does not.

In general, the work is interesting and carefully done, and represents a significant step forward in ESST and molecular manipulation generally. deserves publication, but not in Nature Communications, and not in any journal with its current title.

Comments to the Author

In this contribution Lee, Abdiha, Sartakov, Meijer and Eibenberger-Arias, show that they can nearly completely enantioselectively and quantitatively excite one of the enantiomers of 1-indanol to a particular excited state. This achievement is indeed novel and of great interest and importance to a wide community of chemists and physicists. The manuscript is written in a concise and clear manner. Although my overall impression of this work is favorable, I would like to seek further clarification from the authors regarding the following two points that

- 1) *As a relative outsider, not being involved with this kind of high order microwave spectroscopy, it seems that in general one would want to reach the following two goals (a and b):*
 - a. *Achieve a high enantiomeric purity in the target excited state: of all molecules in the particular target excited state, close to 100% should be of a particular handedness*
 - b. *Quantitative transfer of the population of one particular enantiomer from the ground state to the target excited state: of all molecules in the probe volume, close to a 100% of all molecules with a particular handedness, say R, should be in the target excited state. This second target, introduces a 'novel' additional kind of enantiomeric excited state purity parameter that is different from what chemists and spectroscopists are used to.*

Now my first question to the authors is how they have actually defined their 'enantiomeric enrichment? Here, I refer directly to what seems to be the most important sentences in the paper "These values yield a maximum state-specific enantiomeric of 90.2(1.9)% for the (R)-enantiomer and 92.4(2.1)% for the (S)-enantiomer." on page 9

As stated in our manuscript on page 7, state-specific enantiomeric enrichment refers to the "enantiomeric excess achievable in a chosen rotational state when starting from a racemic mixture." To clarify, we have added the following sentence in the manuscript on page 7 "For example, an 80% state-specific enantiomeric enrichment means that, starting from a racemic mixture, 80% of the molecules in the target rotational state are one enantiomer, while 20% are the racemic mixture. In other words, this corresponds to a composition of 90% of one enantiomer and 10% of the other enantiomer in the target state."

Is this refer the enrichment in one enantiomer relative to all molecules in that excited target state or is it the enrichment relative to all molecules in the probe volume, irrespective of their state?

State-specific enantiomeric enrichment is determined for all molecules in the target state within the probe volume at a certain time. This state can be an excited rotational state as well as the rotational ground state, depending on the choice of the microwave pulse sequence. Enantiomer-specific state transfer is achieved between the two rotational states connected by the final MW pulse. In our study, enantiomeric enrichment is realized in two excited rotational states, but we exclusively probe the population in state $|101\rangle$. To clarify this, we have added an explanation to the manuscript on page 6:

"Note that other pulse sequences can be used for ESST if the first MW pulse drives a transition from the initially populated rotational state. Enantiomer-specific state transfer is then achieved between the two rotational states connected by the final MW pulse. In our chosen sequence, enantiomeric enrichment is realized in two excited rotational states, but we only probe the population in the target rotational state."

In addition, I have difficulty reproducing the numbers stated for the enantiomeric enrichment (“90.2 (1.9)% for the (R)-enantiomer and 92.4(2.1)% for the (S)-enantiomer.” on page 9) from the data in the Figure 3b.

I suggest that in the supporting information the authors detail their calculation of the enrichment.

The values of 90.2(1.9)% for the (R)-enantiomer and 92.4(2.1)% for the (S)-enantiomer stated on page 9 are derived from the data presented in Figure 3b. These percentages are calculated as the ratio of the amplitude to the mean of the sine wave. We acknowledge that this might not have been immediately apparent. To address this, we have added an additional significant digit in the figure and have clarified in the manuscript “The amplitude-to-mean ratio yields a maximum state-specific enantiomeric enrichment of 90.2(1.9)% for the (R)-enantiomer and 92.4(2.1)% for the (S)-enantiomer.”

2) The experiments performed in this study seem to be done only on enantiomerically pure samples of the R and S antipodes of the indanol. Now in an actual experiment on a 50%-50% mixture of R and S molecules, transfer of excitation energy between R and S molecules seems theoretically possible. Such energy transfer would lower the enantiomeric enrichment that can be reached in the actual racemic mixture. Could the authors comment on the likelihood of such energy transfer under the conditions of the experiment?

Transfer of excitation energy between R and S molecules will not occur in the dilute environment of the molecular beam.

Comments to the Author

The paper entitled "Full quantum state control of chiral molecules" reports on enantiomer-specific state transfer (ESST) experiments to achieve almost complete enantiomer-enrichment in a rotational state using rotational spectroscopy combined with LIF. The main result of the current study with respect to previous works by the authors and others is the increase in the enantiopurity of the rotational state, reporting in this case up to a 96%. The authors successfully implemented a depletion scheme including MW-UV double resonance that allows to removal of the thermal population before performing ESST experiments with MW fields. The experiments are competently performed, and the results are sound and interesting. The paper is clearly written and reports the experimental finding in a clear manner. However, I consider that they are a natural extension of previous research and very limited to a very reduced pool of chiral molecules. This is because, the current system fulfills some of the basic requirements that won't be easily found in most chiral molecules, such as having a chromophore group and the ability to work on the rotational ground state 000. As stated by the authors, the use of IR laser could help circumvent the first requirement as most chiral molecules do not have a chromophore group, but even in the case vibrational to rotational energy transfer will potentially affect achievable enrichment.

Vibrational lifetimes are in the millisecond range and vibrational relaxation will basically not occur on the timescale of the molecular beam experiment. Therefore, employing an IR laser instead of UV radiation would not affect the achievable state-specific enantiomeric enrichment and our approach is applicable to all chiral molecules of C_1 symmetry.

The ability to work on the rotational ground is a major limitation. This is because the required MW or RF frequencies connected to the ground state are largely affected by the size and geometry of the system under study, making it in some cases hard to reach. Without fulfillment of this, the M-degeneracy will impact the state transfer, limiting thus the enantiopurity of the target rotational level.

We view the ability to work on the rotational ground state as an advantage rather than a limitation. Our study focuses specifically on the simplest triad that connects to the absolute ground state, which exists for any chiral molecule. While the required frequencies for this simplest triad can vary, current technology can accommodate a broad range of MW frequencies. For instance, the rotational frequencies for 1-indanol in this simplest triad are around 1-3 GHz. Extending this approach to a wider range of chiral molecules, typically heavier than 1-indanol, may require lower frequencies, but also these are readily manageable with existing technology. Although our setup is optimized for the frequencies of 1-indanol, it can be adapted to different frequency ranges with minimal effort.

With this being said, I am hesitant to recommend this paper for publication in Nature Communications. I believe that the novelty of the current study is mostly incremental with respect to the authors' previous work (Physical Review Letters 128, 173001 (2022)), and the general applicability to most chiral molecules is rather limited. Despite the undeniable quality of the work, I find it more appropriate for a more specialized journal.

We respectfully disagree with the referee's assessment that our work is "incremental" for several reasons. Our study goes beyond merely improving efficiency; it unlocks the full potential of the ESST technique. For the first time, our research experimentally establishes the ideal condition required to achieve perfect state-specific enantiomeric enrichment. This breakthrough is not incremental but foundational. Moreover, our work integrates control over the internal quantum states with molecular chirality. This approach allows extending the

research field of state-selected molecular beams to studies on chiral molecules. Thus, we believe our work will contribute broadly to the scientific community's understanding of chiral molecules and their interactions. We hope that our clarifications will alleviate the referee's hesitation to recommend our paper for publication in Nature Communications.

Nevertheless, I have several comments that the authors might want to consider:

1. I find the figures to be clear and very informative. I recommend reporting the frequencies in GHz to improve readability. Also, in the caption of Figure 2 the notation is JKaKc.

We thank the referee for pointing this out. We have addressed the recommendation by changing the frequency units to GHz in Figure 2 and we have corrected the notation to $\mathcal{L}_{K_a K_c}$ in the caption of the Figure 2.

2. Related to this, I was not able to find some important specifications of the amplifiers such as frequency coverage and output power. This info is given for the laser.

We now include the specifications of the amplifiers, including frequency coverage and output power, in the Methods section.

3. There is very little to no information about the generation and control of the pulses during the ESST stage. In a paper presenting an experimental setup, I think this information is relevant and should be included in the methods or SI.

Thank you for pointing this out. We agree that detailed information about the generation and control of the pulses during the ESST stage is important. We added a dedicated section to the Methods giving additional information on microwave pulse generation and control. This will also include the requested information on the microwave hardware employed.

4. I believe that the section about the Rabi frequencies (Rabi flip angles in reality) is introduced without context. A brief introduction or even a small scheme would help the nonspecialist follow the reasoning better.

To enhance clarity for non-specialists, we have included a brief introduction to the concept of Rabi oscillations in the Methods section, along with the Rabi oscillation measurement procedure. We now refer to this addition in the main text on page 19.

5. The authors state that this experiment can be performed in a racemic mixture, however, the authors only report results using enantiopure samples. It would be valuable to show the results using a racemic mixture. This would also avoid using different excitation pulses that complicate the experiment. Under normal conditions, both enantiomers should exhibit the same optimal conditions.

Under our experimental conditions, using a racemic mixture would achieve a 92% enantiomeric excess ($(R-S)/(R+S)$) in the chosen rotational state. However, when a racemic mixture is used, the chirality-sensitive oscillations in the ESST signal as shown in Figure 3b are not observable because the contributions from the (R) and (S) enantiomers cancel each other out. This is why we utilized enantiopure samples to demonstrate our results. As reported by Pérez et al. (*J. Phys. Chem. Lett.* 9, 4539 (2018)), one can measure the achieved state-specific enantiomeric enrichment in a racemic mixture by performing ESST followed by a three-wave mixing experiment. While we acknowledge the value of showing the results with racemic mixtures, our focus in this paper is to demonstrate the full potential of the ESST

technique by experimentally realizing the ideal conditions for enantiomer-specific state transfer.

6. *Related to this, it is unclear why different rotational temperatures were used to calculate normalized ESST curves for each enantiomer. This needs to be further clarified.*

In our study, separate molecular beam sources were used for the two enantiomers to avoid cross-contamination, leading to slight differences in source conditions, including the rotational temperature. These differences do not affect state-specific enantiomeric enrichment, as the two upper levels are depleted prior to ESST, but the rotational temperature does influence the relative scale of the y-axis in Figure 3b. By fitting the rotational temperature using experimental values, we ensure accurate normalization of the ESST curves while preserving enantiomeric enrichment.

7. *Lastly, have the authors considered dephasing from the ESST region to the detection? This could also affect enantiomer enrichment. This should be dependent on the distance between the ESST region and detection.*

Dephasing, which refers to the relaxation of off-diagonal elements of the density matrix to zero, would destroy the coherence, if happening during the ESST process. However, the whole ESST process happens within 4 microseconds and our observations show that coherence is maintained during ESST. In the detection region, we measure the population of the target rotational state, a diagonal element of the density matrix. This means that even if dephasing occurs from ESST to detection, it does not affect the measured enantiomeric enrichment. However, in-beam collisions between molecules and the carrier gas during transit from the ESST region to the detection region can change the population of the target state. To address this, we have accurately modeled these in-beam collisions and accounted for them in our results.

Comments to the Author

This manuscript describes an optimized version of the experiment first reported in Lee et al. (10.1103/PhysRevLett.128.173001). The new experiment improves the initial depletion step of the previous experiment, by adding a microwave field to resonantly couple the 1_{10} and 1_{01} states. This field is switched between two orthogonal polarizations to transfer population from all spatially degenerate levels of 1_{10} to the 1_{01} level where population is optically pumped to an excited electronic state and effectively “removed”. This preparation leaves population only in the initial 0_{00} level of the three-level experimental system. A thus prepared sample of either (S)- or (R)-1-indanol seeded in a neon molecular beam is subjected to the standard three pulse sequence for enantiomer-selective state transfer (ESST) to the final 1_{01} level. Detection is performed by LIF of the same electronic transition used in the depletion step. With optimized π or $\pi/2$ pulses on the ESST step, the authors demonstrate 90% or better enantiomeric enrichment on the basis of the contrast (amplitude/mean ratio) obtained by varying the phase of the final microwave pulse. A phase shift of 180° is observed between the (R)- and (S)-enantiomers as expected and agreement with the theoretical model is excellent. The residual racemic population of the final level is attributed primarily to re-population of the depleted 1_{01} and 1_{10} by in-beam collisions, and experimental evidence and modelling demonstrate that this mechanism is operative. The authors point out that if their technique were to be applied to a racemic mixture, the target 1_{01} state would be composed of up to 96% a single enantiomer. The utility of such enantiopure state preparation for chemical physics and investigations of parity violation in chiral molecules is significant.

I recommend this manuscript for publication after consideration of the minor comments below. The manuscript is well written and contains easily interpretable figures, and the study itself is well-designed, and carefully executed and analyzed. The results represent a major advance in the manipulation of chiral molecules with far-reaching implications. It will be of wide and significant interest for the readership of Nature Communications.

Page 5 Line 22, Page 6 Line 1, Fig.2, Page 14 Lines 7 & 9

In several instances, a quantum number M_z is referred to. I understand this to mean the space-fixed projection of the total angular momentum along the Z-axis. However, the authors also use the label M_J which typically refers to the space-fixed projection of the total angular momentum along the quantization axis, which is typically the Z-axis by convention and is specifically the Z-axis in this experiment. Thus, these two labels to my understanding refer to the same quantity. Perhaps I am being over-simplistic and the authors feel that making the distinction between M_J and M_z is especially important. If that is the case, I would make the difference between the two labels as transparent as possible.

We thank the referee for pointing this out and agree that a consistent notation will be helpful to the reader. We therefore consistently refer to M_J throughout the manuscript and the figures.

Page 10 Lines 8-10

From Equation 1, I see that the signal is controlled by the phases of the three MW fields. Is the \pm distinction known a priori for any pair of enantiomers or can be reliably calculated? I believe that quantum chemistry calculations provide good predictions – this might be specified as an additional requirement.

Yes, the \pm distinction can be reliably determined when the absolute phases of the microwaves are known. We therefore stated in the manuscript: “the handedness of the coordinate system dictates the handedness of the molecules associated with the \pm sign.” For example, if the

coordinate system formed by the three MW fields is right-handed, the + sign corresponds to the (*R*)-enantiomer. Therefore, with the absolute phases of the three MW fields known, it is possible to determine the absolute configuration, even within a racemic mixture.

For my own understanding, is it correct that what distinguishes this method from the MW three-wave mixing technique is that the precise phase relationship of three applied fields must be known here, rather two applied and one detected field (much harder) in the case of MW3M?

In MW three-wave mixing, the enantiomeric excess of a molecular sample can be determined by measuring the phase of the induced field. In the ESST experiment, all three phases are controlled to achieve the optimum transfer. This technique allows the information on enantiomeric excess to be stored in the population of the target level, which is not prone to dephasing, contrary to a measurement of the free induction decay in MW three-wave mixing experiments.

General

It appears that one significant limitation in implementing this approach with a racemic mixture is that, unless the enantiomeric excess of the sample is previously known, the enantiomeric state purity of the racemate must be inferred from an enantiopure study. However, as the authors showed, even subtle differences in the nozzle/source conditions can impact that optimal parameters for ESST. A small change in the optimal relative MW phase and pulse length may lead to a different enantiopurity than what would have been inferred from an enantiopure experiment with optical parameters. Can the authors comment on this limitation? Are there approaches to verify the final enantiopurity through a measurement? I know that in other experiments on enantiomer-selective population enrichment, the final purity can be measured by MW3M, however that would not seem to be compatible with this experiment.

In principle, using enantiopure samples is the simplest way to measure state-specific enantiomeric enrichment. However, the excellent agreement between theory and experiment shown in our study indicates that this is no longer necessary. Under ideal conditions for ESST, one can achieve 92% of state-specific enantiomeric enrichment with any sample, from enantiopure to racemic.

Regarding optimal parameters, using separate sources can lead to slight differences in π -pulse durations, which affects only the relative MW phase, not the state-specific enantiomeric enrichment itself. Therefore, even if the source settings differ, by measuring the Rabi frequencies of all three microwave transitions, one can accurately determine the $\pi/2$ and π conditions for each transition, ensuring optimal state-specific enantiomeric enrichment.

If measuring the state-specific enantiomeric enrichment using a racemic mixture is necessary, MW three-wave mixing following ESST, as reported by Pérez et al. (*J. Phys. Chem. Lett.* 9, 4539 (2018)), can be employed. This can be integrated into our experiment.

Comments to the Author

The authors report a new demonstration of Enantiomer-Specific State Transfer, which yields a high measured transfer yield, due to careful control of microwave fields, and (critically) depletion of several states via optical pumping. Using their own definitions of fidelity, the authors demonstrate the highest ESST to date, at 96 percent. The remaining 4 percent come from small technical imperfections, such as residual collisions within the beam.

This work is carefully done, and the 96 percent purity is convincingly demonstrated. In fact, the abstract accurately summarizes what has been done by the authors. The title does not.

The title of the paper, “Full quantum state control of chiral molecules”, is misleading. Full quantum state control of ANY polyatomic molecule – let alone a chiral one – would be an impressive achievement, well beyond the state of the art. This has not been achieved here. The molecules in the beam start in a thermal distribution of states, at 1.1 K, and none of the steps taken – optical pumping, or reversible microwave pulses – significantly reduce the entropy of this beam. “Full quantum state control” would mean, at a minimum, that the molecules are put in a single quantum state, which is not the case. In addition, the hyperfine states of the molecules are not addressed in any form. A more accurate title would be something like “near unity enantiomer-specific state purity from a racemic sample”. The title obviously carries less punch- but it reflects the work performed, which the current title does not.

In general, the work is interesting and carefully done, and represents a significant step forward in ESST and molecular manipulation generally. Deserves publication, but not in Nature Communications, and not in any journal with its current title.

It is non-trivial to create a title that fully captures the essence of a paper. The title of a paper has to be read in combination with the abstract, at least, to get a proper understanding on what has been achieved. We acknowledge that “Full quantum state control of chiral molecules” may imply a broader achievement than our study demonstrates. If the referee insists on a different title, we are willing to change it to “Complete chiral separation at the quantum level”.

Reviewer #1 (Remarks to the Author):

revision OK. Publish without further revision

Reviewer #2 (Remarks to the Author):

I appreciate the author's efforts to address some of the issues raised by the other reviewers and myself. However, the primary obstacle to a positive recommendation for publication in Nat Comm remains. As previously stated in my initial review, this paper does not present any fundamentally novel findings in comparison to previous research (Physical Review Letters 128, 173001 (2022)). The current work can be classified as either incremental—an improvement with respect to a previous work (the one with the actual innovation)—or as a “breakthrough and foundational,” as the authors state. I believe this work belongs to the former category, and there is an attempt to publish a work in a high-impact journal without the actual impact. The ideal conditions were already established, and the present work is simply an extension of them. The work is well performed, and the results are of good quality. However, the novelty and impact were already reported by the authors, and therefore I recommend publication in a more specific journal.

Some other points:

1-I was not referring to vibrational lifetimes, but to vibrational to rotational energy transfer (V-R energy transfer), which refers to the transfer of energy initially present in vibrational modes to rotational modes. As the complexity of the molecule increases, this can be relevant and affect the achieved enrichment.

2-It is evident that the ability to work on the rotational ground state represents a significant advantage. This is the primary focus of the paper. However, the challenge lies in the ability to do so due to technical restrictions. As the size of the molecules increases, which is relevant for the application of the current technique, it becomes increasingly difficult to work on the ground state. Some transitions may fall within the radio frequency (RF) regime, where the available power and efficiency of broadcasting may be suboptimal. This could necessitate the use of longer excitation pulses to achieve optimal Rabi oscillations, which may limit the applicability of the experiment.

3-The authors mention the approach reported in J. Phys. Chem. Lett. 9, 4539 (2018) as a possible way to observe the enantiomeric excess when using a racemic mixture, thus this citation needs to be included.

4-I agree with the overselling character of the title.

Reviewer #3 (Remarks to the Author):

The authors have satisfactorily responded to all questions and comments. The changes and additions to the manuscript improve the clarity for both expert and broad readership alike. I recommend this manuscript in its revised form for publication in Nature Communications.

Comments to the Author

I appreciate the author's efforts to address some of the issues raised by the other reviewers and myself. However, the primary obstacle to a positive recommendation for publication in Nat Comm remains. As previously stated in my initial review, this paper does not present any fundamentally novel findings in comparison to previous research (Physical Review Letters 128, 173001 (2022)) The current work can be classified as either incremental—an improvement with respect to a previous work (the one with the actual innovation)—or as a "breakthrough and foundational," as the authors state. I believe this work belongs to the former category, and there is an attempt to publish a work in a high-impact journal without the actual impact. The ideal conditions were already established, and the present work is simply an extension of them. The work is well performed, and the results are of good quality. However, the novelty and impact were already reported by the authors, and therefore I recommend publication in a more specific journal. Some other points:

1-I was not referring to vibrational lifetimes, but to vibrational to rotational energy transfer (V-R energy transfer), which refers to the transfer of energy initially present in vibrational modes to rotational modes. As the complexity of the molecule increases, this can be relevant and affect the achieved enrichment.

There appears to be a misunderstanding here; Even when IR excitation would be used for depletion (and detection), the ESST process would still be applied to the vibrational ground state. Here, vibrational to rotational energy transfer can obviously not occur.

2-It is evident that the ability to work on the rotational ground state represents a significant advantage. This is the primary focus of the paper. However, the challenge lies in the ability to do so due to technical restrictions. As the size of the molecules increases, which is relevant for the application of the current technique, it becomes increasingly difficult to work on the ground state. Some transitions may fall within the radio frequency (RF) regime, where the available power and efficiency of broadcasting may be suboptimal. This could necessitate the use of longer excitation pulses to achieve optimal Rabi oscillations, which may limit the applicability of the experiment.

While the required frequencies for rotational transitions from the ground state can vary, current technology can accommodate a broad range of MW and RF frequencies. The required use of RF radiation in the case of large molecules might demand different technical solutions. For example, instead of horn antennas as we have used now for the MW radiation, transmission lines, in which the electric field is concentrated between two metallic strips, can then be used.

3-The authors mention the approach reported in J. Phys. Chem. Lett. 9, 4539 (2018) as a possible way to observe the enantiomeric excess when using a racemic mixture, thus this citation needs to be included.

This reference has been added on page 10: "The data presented here show near-complete quantum state control of the chiral molecule 1-indanol to the extent that more than 96% enantiomer-selectivity can be obtained when starting from a racemic mixture, which can be verified following a scheme presented elsewhere³³"

4-I agree with the overselling character of the title.

Our newly proposed title is "Near-complete chiral separation in quantum states"